# Proteins Involved in Focal Cell Adhesion and Podosome Formation Are Differentially Expressed during Colorectal Tumorigenesis in AOM-Treated Rats

**DOI:** 10.3390/cancers16091678

**Published:** 2024-04-26

**Authors:** Ian X. Swain, Adam M. Kresak

**Affiliations:** Department of Pathology, School of Medicine, Case Western Reserve University, 2103 Cornell Road, Cleveland, OH 44106, USA; adam.kresak@uhhospitals.org

**Keywords:** colorectal tumorigenesis, cancer, regulatory proteins, azoxymethane, aberrant crypt, adenoma

## Abstract

**Simple Summary:**

Colon cancer first involves the development of small clusters of abnormal cells. The purpose of our study was to determine changes in protein expression in such abnormal cells as they grow after being triggered to form due to exposure to a carcinogen using rats as a model. We were able to identify a number of proteins that were very differently expressed when comparing early abnormal cells to more mature cell clusters. Many of these proteins are involved in regulating cell-to-cell interactions and cell shape, among other activities in the cells. Our findings help us better understand the biology of intestinal cancer and may be useful in developing biomarkers for detecting colon cancer. This information also helps us lay a foundation for future studies aimed at better understanding the how intestinal cancers develop and grow under different intestinal conditions.

**Abstract:**

Colorectal tumorigenesis involves the development of aberrant crypt foci (ACF) or preneoplastic lesions, representing the earliest morphological lesion visible in colon cancer. The purpose of this study was to determine changes in protein expression in carcinogen-induced ACF as they mature and transform into adenomas. Protein expression profiles of azoxymethane (AOM)-induced F344 rat colon ACF and adenomas were compared at four time points, 4 (control), 8, 16, and 24 weeks post AOM administration (*n* = 9/group), with time points correlating with induction and transformation events. At each time point, micro-dissected ACF and/or adenoma tissues were analyzed across multiple quantitative two-dimensional (2D-DIGE) gels using a Cy-dye labeling technique and a pooled internal standard to quantify expression changes with statistical confidence. Western blot and subsequent network pathway mapping were used to confirm and elucidate differentially expressed (*p* ≤ 0.05) proteins, including changes in vinculin (*Vcl*; *p* = 0.007), scinderin (*Scin*; *p* = 0.02), and profilin (*Pfn1*; *p* = 0.01), By determining protein expression changes in ACF as they mature and transform into adenomas, a “baseline” of altered regulatory proteins associated with adenocarcinoma development in this model has been elucidated. These data will enable future studies aimed at biomarker identification and understanding the molecular biology of intestinal tumorigenesis and adenocarcinoma maturation under varying intestinal conditions.

## 1. Introduction

Cancer detection methods have advanced greatly during the past few decades. However, colon cancer is one of the top three causes of cancer-related mortality today [1]. Although early detection is known to be positively associated with favorable prognosis and long-term survival, colorectal cancer is most often discovered after a patient has crossed the clinical horizon (symptoms appear) and at a stage when metastatic cancer cells have spread [2]. The pathogenesis of colorectal cancer remains poorly understood. Identifying novel markers, predictive of neoplastic changes and tumorigenesis, is crucial in reducing onset of the disease.

Colorectal tumorigenesis begins with transformation of normal crypt cells into preneoplastic lesions or aberrant crypt foci (ACF). ACF may mature, develop, and transform into adenomas (non-cancerous tumors) before becoming carcinomas [3]. The protracted development and location of ACF and early adenomas, the obvious challenges of accessing samples, and risk associated with even routine biopsies make humans a problematic model for conducting studies on intestinal tumorigenesis. Hence, a chemically induced murine (rat) model has often been used to investigate prevention and pathogenesis of this disease.

Azoxymethane (AOM) is a chemical carcinogen that has been used to produce pre-neoplastic ACF and, subsequently, adenomas (tumors) in the colons of rodents [3,4,5]. ACF have also been detected in the human colon [6] and studies in both humans and murine models have shown that colonic ACF are precursor lesions from which adenocarcinomas develop [7,8,9,10,11,12]. This characteristic of ACF makes them a highly useful target for investigating markers of human colon carcinogenesis. In addition to morphological similarities, the molecular characteristics of ACF in rats also show numerous similarities to human colon cancer, with analogous features including K-ras [4] and β-catenin mutations [5], microsatellite instability [12], overexpression of cyclooxygenase-2 [13,14], altered mucin secretion [15,16], and changes in the activity of enzymes involved in modulation of oxidative stress [17,18,19]. These shared metabolic features in rats and humans are associated with an increased risk of colorectal tumorigenesis, making the AOM rat model a good approach for studying ACF maturation and transformation into adenocarcinomas.

Cancer cell biology studies from breast [20], prostate [21], ovarian [22], and colon [23] cancers typically identify hundreds of differentially expressed regulatory proteins. However, these differentially expressed proteins may or may not be relevant to carcinogenesis and tumor development and maturation. The purpose of this study was to identify differentially expressed regulatory proteins relevant to ACF maturation and transformation into intestinal adenocarcinomas and establish baseline AOM-induced ACF molecular/protein expression characteristics. These data will enable future studies aimed at biomarker identification and understanding the molecular biology of intestinal tumorigenesis and adenocarcinoma maturation under varying intestinal conditions.

## 2. Materials and Methods

### 2.1. Reagents

The majority of the chemicals used in this study were obtained from GE Healthcare (Piscataway, NJ, USA), Thermo Scientific (Rockford, IL, USA), and Invitrogen (Carlsbad, CA, USA), without further purification, unless otherwise stated. Azoxymethane (AOM) was obtained from (Sigma-Aldrich, St. Louis, MO, USA). Antibodies to ACY-1 (Sigma-Aldrich), Vcl, and Pfn1 (Cell Signaling Technology, Danvers, MA, USA) were purchased from the indicated vendors.

### 2.2. Animal Preparation and Carcinogen Treatment

Thirty six, four-week-old, weanling, male F344 rats (Charles River, Kingston, MA, USA) were housed individually in microisolator cages in a room controlled for temperature (21 ± 1 °C), humidity, and light (12 h light/dark cycle). Rats were allowed 7 d for acclimatization before intraperitoneal (i.p.) injection with the colonotropic carcinogen AOM (high purity, Sigma-Aldrich) at 20 mg/kg body wt in 0.9% NaCl, once weekly for 2 weeks, as described by Pierre et al. [16], then sacrificed 4 (control), 8, 16, and 24 weeks (*n* = 9/group) after the first AOM injection to harvest ACF, adenoma, and colon tissues (Figure 1 shows the study design).

Rats consumed water and standard diet (AIN-93G diet [24], Cat.# TD.94045; Harlan Teklad Laboratories, Madison, WI, USA) ad libitum (Table 1 shows the diet composition). All animal procedures followed the Institutional Animal Care and Use Committee procedures at Case Western Reserve University, in accordance with NIH guidelines.

### 2.3. Tissue Collection

At each sampling, rats were selected randomly and then selected rats were fasted overnight before tissue collection. Rats were sacrificed by cardiac puncture following i.p. administration of sodium pentobarbital (50 mg/kg body weight). Intestines were removed and colons excised from the caecum, at the ileocecal valve to the anus, and rinsed with ice-cold sucrose/Tris (0.25 M sucrose/10 mM Tris, pH 7.4) solution to remove any debris. Colon length measurements were recorded and the colon was opened longitudinally on an ice-cooled, sterile platform. ACF and/or adenomas (classified as supra-ACF growths in excess of 0.25 mm diameter) were counted, the dimensions were measured, and then they were excised (“plucked”) using a sterile micro-dissection scalpel and fine forceps under 20–40× magnification using a Leica M125 C stereo microscope (Leica Microsystems, Buffalo Grove, IL, USA), while resting on filter paper above a bed of crushed ice. ACF and/or adenomas (polyps) were excised from colons, then immediately frozen using liquid nitrogen until processed for use. All ACF and adenomas visible under 20–40× magnification from the entire colon of each animal were collected, pooled, and homogenized for use in assays, with nine replicates per group (for each of the time point samplings).

Tissues harvested for protein expression analysis were not stained in order to prevent interference with the assays. The control was the T1 (wk 4) time point. Because this study involved comparisons of the protein profiles of ACF as they mature and transform into intestinal (colorectal) adenomas, adjacent normal (non-ACF and/or non-adenoma) tissues were not collected. Only ACF that were distinguished from the surrounding normal tissue, identified by their slit-like opening, increased size, and peri-cryptal zone as a border-like region of differentiation, as previously noted [4,6], visible at 20–40× magnification (Leica M125 C stereo microscope; Leica Microsystems, USA), and in excess of 0.25 mm diameter were collected. ACF and adenomas that were clearly visible as small, tumor/polyp-like projections rising above the surrounding mucosa were collected. The T4 time point of 24 weeks was selected based on previous research that showed that carcinogen-induced adenomas are usually seen starting at ~5 months [6,9]. ACF and adenoma number and sizes were measured at each time point. To verify and confirm that ACF and adenoma were collected, a 0.75 cm section of the distal colon of two rats from the first sampling was taken and ACF were determined according to the method described by Bird using methylene blue stain [3]. Tissues were examined microscopically after H&E staining by a pathologist experienced in visualization and identification of ACF and adenomas. To confirm adenomas were not lymphoid aggregates, tissue samples were also randomly selected for confirmational analysis by the CWRU Histology Core Center.

### 2.4. Protein Extraction and Fluorescence Labeling

Protein extraction and fluorescence labeling were performed as previously described [25], with some modifications. Briefly, total protein was extracted from the respective samplings of pooled ACF and adenomas from each animal, at each time point, using a mortar and pestle to grind the frozen tissue samples while in liquid nitrogen into a fine powder, which then was homogenized in lysis buffer (2 M thiourea, 7 M urea, 4% (*w*/*v*) CHAPS, 30 mM Tris) with four cycles of freeze, thaw, and sonication. Also, a dye-swapping approach was used so that the nine replicate samples for each time point sampling were variously labeled with Cy3 or Cy5 to control for dye-specific labeling artifacts. Labeling was performed by incubating for 30 min on ice under dark conditions, the reaction was quenched with 10 mM lysine, and samples were incubated for another 10 min. Each gel contained the pooled standard (equal aliquots of all the samples in all groups) and two other subject (time point) samples. Protein sample size for this study was estimated from power analysis, as performed in previous studies [26,27].

### 2.5. Gel Electrophoresis

Gel electrophoresis was performed as previously described [25], with adjustments. Briefly, before performing SDS-PAGE, strips were equilibrated with 10 mL equilibration buffer A (100 mM Tris-HCl, 8 M urea, pH 6.8, 30% (*v*/*v*) glycerol, 1% (*w*/*v*) SDS, 5 mg/mL DTT) before addition of 10 mL equilibration buffer B (100 mM Tris-HCl, 8 M urea, pH 6.8, 30% (*v*/*v*) glycerol, 1% (*w*/*v*) SDS, 45 mg/mL iodoacetamide) for an additional 10 min to help prevent point streaking [28]. Strips were then run on 12.5% acrylamide isocratic Laemmli gels [29]. Electrophoresis was also performed in a dark environment.

### 2.6. Image Acquisition and Protein Quantification

Image acquisition and protein quantification were performed as previously described [25], with modifications. Briefly, gels were scanned using a 488 nm laser with an emission filter at 520 nm band pass (BP) 40, a 532 nm laser with emission filter at 580 nm BP 30, and 633 nm laser with emission filter at 670 nm BP 30 to obtain the Cy2, Cy3, and Cy5 images, respectively. Data analysis was carried out via gel images from four time points (*n* = 9 per time point sampling), with each gel including internal standards, which were pooled mixtures of equal aliquots of each experimental sample. Sample (Cy3 or Cy5 labeled) spot maps were assigned into four groups, with each group representing 4 (the control), 8, 16, and 24 weeks after AOM-mediated colonotropic carcinogen induction, respectively.

### 2.7. In-Gel Digestion and Protein Identification

In-gel digestion and protein identification were performed as previously described [25], with updates to the method. Briefly, using the International Protein Index (IPI) rat database (now SwissProt), results were used to interrogate sequences present in IPI rat via Mascot version 2.5 (Matrix Science, London, UK). Mascot searches were performed with a maximum peptide and fragment ion mass tolerance of 10 ppm and 0.8 Da, respectively, variable methionine oxidation, and carbamidomethylation of cysteine. 

### 2.8. Western Blotting

Immunoblotting was performed as previously described [30], with modifications. Briefly, tissue lysates were prepared from rat colon ACF and/or adenomas obtained from two sets of biological replicate animals for each time point sampling. Membranes were blocked with 5% skim milk or 5% BSA for 1 h at room temperature with gentle shaking, washed, then incubated overnight with specific primary antibodies against Acy1 (0.5 ng/µL), profilin-1 (0.130 ng/µL), Vcl, and β-actin (1.000 ng/µL). Antibodies to ACY-1 (Sigma-Aldrich), Vcl, and Pfn1 (Cell Signaling Technology) were obtained from the vendors indicated. Ultimately, bands on the films were scanned and quantified using ImageQuant-TL 7.0 software (GE Healthcare).

### 2.9. Network Analysis

Data were further analyzed using Ingenuity Pathway Analysis software (Ingenuity Systems, Inc., Little Elm, TX, USA) to identify biochemical regulatory pathways and relationships, including direct protein interactions, transcriptional regulation, enzyme–substrate, and other structural and/or functional relationships. Interaction networks were then visualized graphically, with nodes (proteins) and edges (protein relationships) alongside the empirical expression patterns.

### 2.10. Statistical Analyses

Power analysis was performed based on previous published data [23]. Data were analyzed using SAS statistical software (SAS Version 10.2, SAS Institute, Cary, NC, USA). Tukey’s multiple comparisons test was used to differentiate among means. Pearson correlations were used to determine associations between protein expression changes and ACF maturation and transformation into adenomas. Values are reported as the mean ± SD for immunoblotting and mean ± SEM for other results. Differences were considered significant if *p* ≤ 0.05. Illustration of data and results was performed using GraphPad Prism (Software version 10.2; GraphPad, Boston, MA, USA).

## 3. Results

### 3.1. Quantitative Data for ACF and Adenomas

As the time point samplings proceeded from T1 to T4, the pooled samples went from containing solely (100%) ACF (with at least 3 crypts—both dysplastic and non-dysplastic—per focus) to increasingly a mix of ACF and adenoma by wk 3 and 4, although ACF predominated at each time point sampling, including T4 (with approximately 69% ACF tissue). From T1 to T3, the number of ACF increased progressively, then decreased at T4, However, ACF size was greatest at T4 (the numbers of ACF (mean ± SEM) seen in T1–T4 were 32 ± 14.8, 39 ± 19.7, 72 ± 21.4, and 46 ± 17.1, respectively, with sizes (mm) of 0.1 ± 0.02, 0.1 ± 0.05, 0.15 ± 0.04, and 0.17 ± 0.07, respectively). At T1, no adenomas were found. At T2, the first adenomas were seen, though smaller in size. From T2 to T4, the number and size of adenomas increased (the numbers of adenomas (mean ± SEM) seen in T1–T4 time points were 0, 5 ± 1.2, 14 ± 4.5, and 21 ± 5.1, respectively, with sizes (mm) of 0, 0.3 ± 0.07, 0.9 ± 0.2, and 1.1 ± 0.3, respectively). Although the size of adenomas generally increased from T2 to T4, our results did not conclusively represent a progression of ACF to adenoma. Figure 2A–F shows histological imaging.

### 3.2. Comparative Protein Expression Analysis

Analysis of ACF and adenoma intestinal tissues at the four time points (*n* = 9/group) revealed profiles of differentially expressed proteins (Figure 3A–C shows a 2D-DIGE gel overlay, subset gels, and the associated 2D (deep purple-labeled) gel map). On the 2D map, the pick locations of proteins with significantly different expression based on the variance of the mean change among the groups (*p* ≤ 0.05), for the T4 vs. T1 tissue time point comparison, are shown in Figure 4A,B. In some cases, the same protein was identified in different spots across the 2D gel, suggesting the occurrence of post-translational modifications. Appendix A accompanies this paper and data can be found at https://proteomecentral.proteomexchange.org/cgi/GetDataset?ID=PXD051252, accessed on 8 April 2024).

### 3.3. Differentially Expressed Proteins

FT-MS/MS analysis on a total of 68 spots of interest successfully derived sequence data from 60 spots; the digest for 1 spot did not yield a protein ID and the digests for 7 spots did not reveal any peptides. Some spots contained isoforms of proteins due to either post-translational modification or proteolysis. As the overall expression patterns of the observed isoforms were similar, duplicates were omitted. For 25 spots, no significant difference was found between any two time points in at least one time point comparison. Thus, we report and describe in detail the identification of a total of 35 unique proteins with significant (*p* ≤ 0.05) fold-changes in at least one time point comparison (Table 2). 

When comparing the two distal time points (T1 to T4), key insights from our data included significant (*p* ≤ 0.05) changes in Vcl, which progressively decreased (~2-fold); Scin (or adseverin) decreasing by half, Pfn1 decreasing ~2-fold, Acads increasing by approximately 50%, Gsn Iso-2 increasing by nearly 50%, and Cps1 decreasing by approximately 65%. When comparing the two distal time points in reverse (T4 vs. T1), key insights from our data included significant (*p* ≤ 0.05) changes in Eno1 (decrease of 89%), Oat decreasing by 71%, Inpp1 exhibiting a subtle, yet significant decrease of approximately 15%, Prdx4 decreasing by 71%, and Fabp6 decreasing by approximately 60%.

### 3.4. Validation by Immunoblotting

To evaluate changes in protein expression, select proteins identified by network analysis were used for semiquantitative immunoblotting (Figure 5A–E), including Pfn1 (Profilin-1), Acy1 (Aminoacylase-1A), and Vcl, plus met-Vcl (Vinculin, plus metavinculin) to elucidate progressive changes occurring as ACF develop into adenomas. Relative fold-changes for these proteins were determined as described for Table 2. Pfn1 is thought to regulate actin polymerization in response to extracellular signals. Inhibition of Pfn2 has been shown to contribute to colorectal cancer metastatic and migratory capacities [31]. Acy1 is a cytosolic enzyme that catalyzes the hydrolysis of acylated L-amino acids to L-amino acids and acyl group, and its activity has been shown to decrease in small cell lung cancer cell lines and tumors [32]. Vcl is a cytoskeletal protein associated with cell–cell and cell–matrix junctions. Its signaling has been shown to be critical for the maintenance of cell adhesion; Vcl anchors F-actin to the membrane and data in the literature suggest its activity plays a role in cancer cell migration [33]. To gain additional perspective, also included was metavinculin (met-Vcl), a Vcl splice isoform, containing an additional exon, which is selectively expressed at sub-stoichiometric amounts relative to Vcl [34]. Pfn1, Acy1, and Vcl (and met-Vcl) showed generally progressive decreases in expression from the T1 to T4 time points. The immunoblot results were consistent with the network analysis findings and were consistent with the 2D DIGE results. Equal loading of the protein samples was demonstrated and confirmed by re-probing the blots for β-actin (also shown in Figure 5A).

### 3.5. Network Analysis

To better understand our differential protein expression data and possible significance to intestinal ACF maturation and adenoma development, and to ascertain regulatory interactions between other proteins and regulators in known gastrointestinal cancer networks, the differentially expressed proteins we identified were analyzed using the Metacore/Integrated Pathway Analysis (IPA; Version 8.0; Qiagen, Redwood City, CA, USA) tool for multi-OMICs data (Clarivate, London, UK), as previously described [35].

Of the total 35 proteins imported into the MetaCore suite, all 35 were successfully mapped to the MetaCore database. A network of these proteins was then generated, as previously described [36]. Protein–protein interactions (PPIs) and related gastrointestinal tract cancer networks among the differentially expressed proteins identified by 2D-DIGE/MS and proteins from the MetaCore database, when comparing maturing adenoma back to ACF, are shown in Figure 6. Individual proteins are represented as nodes, and the different shapes of the nodes represent the functional classes of the proteins. The edges define the relationships of the nodes, and the arrowheads indicate the direction of the interaction. The color of the node indicates activation (green), inhibition (red), and unspecified (clear grey) interactions. Key proteins found in this study to be differentially expressed and involved in gastrointestinal cancer networks are labeled.

## 4. Discussion

Our findings show there are significant changes in the expression of regulatory proteins in AOM-induced ACF as they mature and develop into intestinal adenomas. Key findings of this study describe the nature of the differentially expressed proteins, which include proteins involved in focal cell adhesion and cell migration, actin polymerization, amino acid catabolism, podosome formation, mitochondrial energy modulation, and regulation of intracellular micronutrient metabolism.

In this study, we also found that expression of vinculin (Vcl; a 117-kDa cytoskeletal protein known to play a role in cell shape control-based focal adhesion structure and function) progressively decreased from the T1 to T4 time points. Vcl signaling has been shown to be necessary for the maintenance of cell adhesion via anchoring F-actin to the membrane [33] and thus, the decrease in Vcl expression we found at later time points in this study may plausibly enable regulation of the formation of maturing adenomas via the lack of membrane anchoring function. Metavinculin (met-Vcl), a splice isoform of Vcl containing an additional exon encoding a 68-residue insert within the actin binding tail domain, has been found to be selectively expressed at sub-stoichiometric amounts relative to Vcl in smooth and cardiac muscle cells [34], enabling membrane flexibility in such tissues. The decrease in expression we observed for met-Vcl from T1 to T4 further support cell membrane changes occurring in developing adenomas.

The activity of scinderin (Scin—or adseverin; a member of the gelsolin family that regulates the actin cytoskeleton by severing pre-existing actin filaments, capping filaments ends, and nucleating actin assembly) in the later time point was surprising, yet plausible. As an actin severing protein, rapidly dividing cells demonstrate decreased expression levels of Scin, as tumorigenic cells have a tendency to migrate, and other studies have also shown that the expression of Scin, like aminoacylase-1A (Acy1; active in colorectal cancer cell migration) is associated with cell proliferation changes [37] and colon cancer survival in humans [38,39]. Interestingly, Scin has also been described by its ability to influence cell differentiation and apoptosis in megakaryoblastic leukemia cells [40].

Our results also concur with other studies that have investigated the role of profilin-1 (Pfn1; involved in regulating actin polymerization in response to extracellular signals). The effects of Pfn1 remain relatively understudied in the area of colorectal cancer. Evidence suggests that expression of its counterpart, profilin-2 (Pfn2), is related to intestinal micronutrient mineral regulation [41]. Loss of or decreased activity of Pfn2 has been correlated with colorectal cancer cell metastatic and migratory capacities [31], and knockdown of a Pfn1-interacting protein regulates the advancement of colorectal tumors [42]. We observed a similar general progressive decrease from T1 to T4 in the expression of aminoacylase-1 (Acy1; a cytosolic, homodimeric, zinc-binding enzyme that catalyzes the hydrolysis of acylated L-amino acids to L-amino acids and acyl groups), which is postulated to function in the catabolism and salvage of acylated amino acids, and its expression has been reported to be reduced or undetectable in small cell lung cancer cell lines and tumors [32].

New observations seen in our data included expression changes of acyl CoA dehydrogenase (Acads; involved in metabolism of fatty acids). Acads catalyzes the dehydrogenation of esters involved in energy production, a feature altered in some types of cancer that require increased energy to proliferate and migrate. However, the association between Acads and intestinal and colorectal cancers has not been widely explored. Some studies have found that Acads are downregulated in cancer cells and expression levels are correlated with clinical cancer stage and predict patient prognosis [43], but the different expression change in Acads observed in our study may be related to the benign properties of the relatively young adenomas we analyzed at the later time points in this study. However, our data are in agreement with a report that showed a positive correlation between the expression of Acads and malignant esophageal cancer [44]. Other novel findings seen in our data were differences in the activities of alpha-enolase, ornithine aminotransferase, inositol polyphosphate-1-phosphatase, and peroxiredoxin-4. Alpha-enolase (Eno1; an important enzymes in glucose metabolism) has sometimes been shown to be upregulated in tumor tissues and has been found to be inversely correlated with regulatory tumor suppressors [45]. However, other studies have found that inhibition of Eno1 leads to altered cell migration and proliferation [46]. Ornithine aminotransferase (Oat; an enzyme that produces glutamate via ornithine) has been shown to be upregulated in hepatocellular carcinomas but has different activity in the intestine, where it produces citrulline and arginine instead of glutamate [47]. This may reflect a tissue-specific response or relate to the sampling in this study of benign earlier stage adenomas at the last (T4) time point, before adenomas become carcinomas. The expression of inositol polyphosphate-1-phosphatase (Inpp1; responsible for dephosphorylating free inositol) exhibited a more subtle change. Although Inpp1 has not been widely investigated in the context of gastrointestinal cancers, it has been shown to be involved in regulation of cancer cell activity via lipid lysophosphatidic acid signaling and thereby, impact cell migration and growth [48]. The expression of peroxiredoxin-4 (Prdx4; a protein involved in elimination of reactive oxygen species) has been associated with cancer cell stage in lung, breast, and gastrointestinal tract cancers [49,50].

Our findings are also in agreement with studies of progressive cancer cell membrane alterations, oxidative adaptions, and altered cancer cell mitochondrial activity. Gelsolin (Gsn Iso-2; an actin regulator that inhibits apoptosis and plays a role in podosome formation) is known to induce cancer cell invasion by altering levels of reactive oxygen species [51]. Also, Gsn has been shown to downregulate expression levels of important cell adhesion molecules [52]. Further, increased expression of Gsn is associated with later-stage cancer [53], which may indicate an increased invasion ability for cancer cells. The expression of carbamoyl-phosphate synthase [ammonia], mitochondrial (Cps1; involved in mitochondrial energy pathways), has been shown to be inversely correlated with tumor development in the context of gastrointestinal cancer and its downregulation is linked to shorter patient survival, and, in intestinal-type cancers specifically, it is associated with advanced tumor stages [54]. Our findings also concur with previous research showing Cps1 is associated with apoptosis in colorectal tumors and later-stage colon cancer tissues possessing decreased Cps1 expression levels [55]. One striking expression difference was based on the greater initial expression seen with gastrotropin (Fabp6; involved binding of long-chain fatty acids and bile acids). The expression of Fabp6 and other genes in its family has been found to be associated with colorectal cancer, and its expression is strongly correlated with cell migration and the immune response [56,57,58].

One limitation of this study may be that 2D-DIGE/MS analyses were not performed on normal (non-ACF or non-adenoma) intestinal tissues. However, the design of this study was to investigate and compare only protein expression changes in ACF as they mature and transform into adenomas, not to compare ACF and/or adenoma to normal tissues. Also, if we expanded our analyses to include normal tissues, then we would not have had the resources to perform the type and number of analyses as performed and use the number of animals per group as we did in this study in order to provide greater statistical power, hence the reason for our statistically strong differential protein expression findings between time point samplings. To reiterate, although the size of adenomas generally increased from T2 to T4, our results do not conclusively represent a progression of ACF to adenoma.

In this study, we were able to successfully map all differentially expressed proteins shown to change (*p* ≤ 0.05) during ACF maturation and transformation into adenomas to biological pathways (protein–protein and other regulatory interactions) related to gastrointestinal tract cancer networks, also visualizing the relationship between many proteins, such as Vcl, Pfn1, Scin, and Gsn, involved in the molecular biology of cancer cells. Our findings provide knowledge and insight into the activities of such proteins in ACF as they mature and develop into adenomas. Our data also add to the understanding of prior studies describing the activity of regulators identified in cancer development pathways in a variety of cancer tissues [59]. We also elucidated new aspects of regulatory protein expression during intestinal tumorigenesis and adenocarcinoma development.

## 5. Conclusions

Our findings enhance the understanding of the molecular biology of intestinal tumorigenesis. These finding may be useful in developing biomarkers for colorectal cancer for use in early-stage detection and help lay a foundation for future studies aimed at better understanding the molecular biology of intestinal tumorigenesis and adenocarcinoma maturation under varying intestinal conditions.

## Figures and Tables

**Figure 1 cancers-16-01678-f001:**
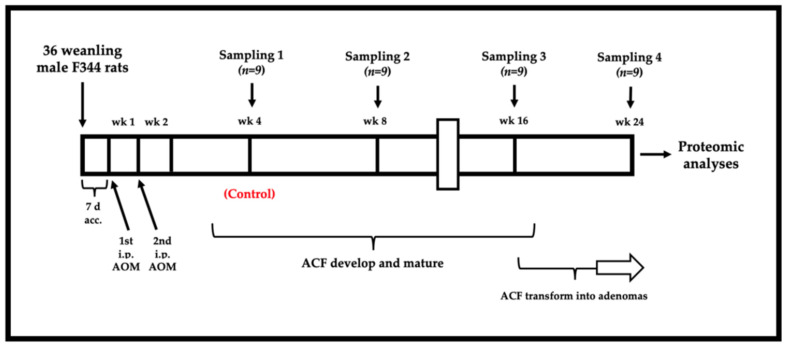
Study design and time point sampling overview. Rats were allowed 7 days for acclimatization (acc.) before intraperitoneal (i.p.) injections at the beginning of weeks 1 and 2 with the colonotropic carcinogen azoxymethane (AOM).

**Figure 2 cancers-16-01678-f002:**
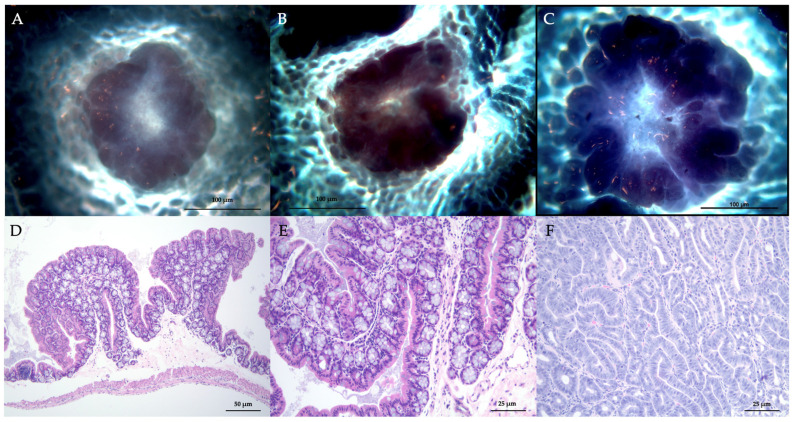
(**A**–**C**) Images of different adenomas at progressive stages of development at T2–T4 (**A**–**C**), respectively, stained with methylene blue; images are backlit. (**D**–**F**) H&E-stained adenoma cross-sections: (**D**) Cross-section of adenoma with bifurcated appearance, (**E**) Cross-sectional close-up near base, and (**F**) Cross-section interior, top–down through adenoma.

**Figure 3 cancers-16-01678-f003:**
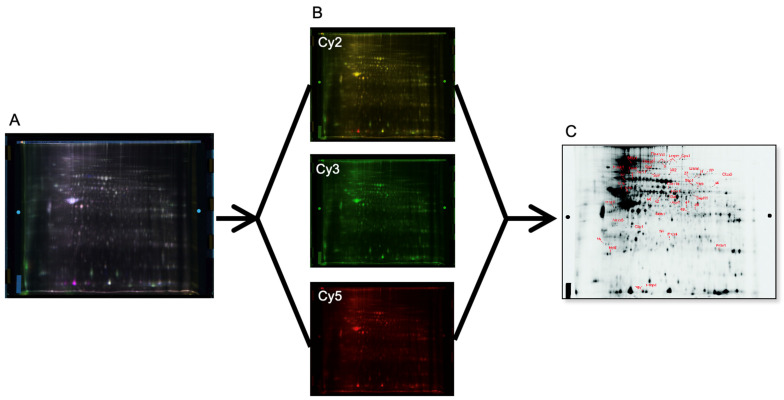
(**A**) Overlay of 2D-DIGE fluorescence gel images, (**B**) individual fluorescence gel images acquired using excitation/emission wavelengths of 488/520 nm, 532/580 nm, and 633/670 nm for Cy2 (internal standard), Cy3 (T1 time point tissue sampling), and Cy5, (T4 time point tissue sampling), respectively, and (**C**) the accompanying 2D map of deep purple-labeled intestinal tissue proteins, showing the pick locations of the subset protein expression changes mediated by azoxymethane.

**Figure 4 cancers-16-01678-f004:**
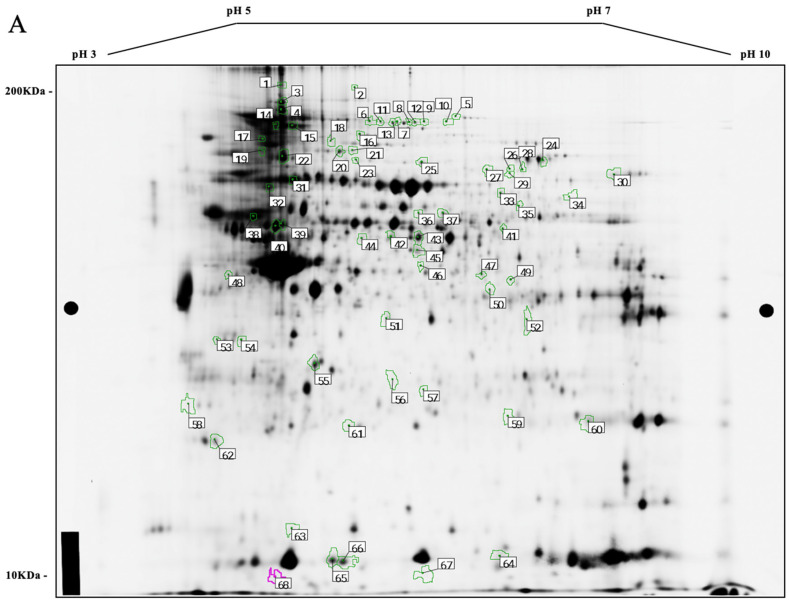
(**A**) Two-dimensional map of deep purple-labeled intestinal cell proteins, indicating pick locations of a subset of proteins that changed as aberrant crypt foci matured and transformed into adenomas, from the T4 vs. T1 time point tissue sample comparison, and (**B**) the same 2D gel (from Figure 3C) showing protein identities associated with spots of significant interest. Orientation on both gels of the pH gradients is shown on the horizontal axes, from pH 3 (**left**) to 10 (**right**), with approximate molecular mass ranges indicated along the vertical axes from 10 (near **bottom**) to 200 kDa (near **top**).

**Figure 5 cancers-16-01678-f005:**
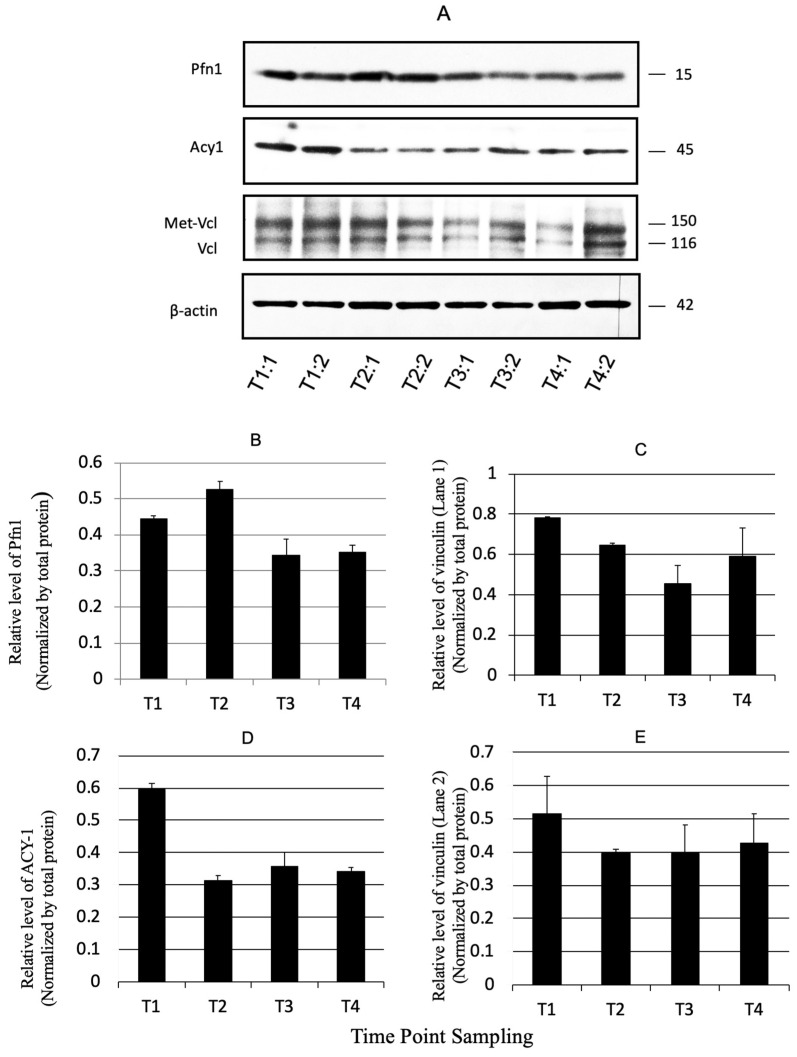
(**A**) Confirmational immunoblots for select proteins, identified as differentially expressed (*p* ≤ 0.05) by 2D-DIGE. The blots were probed with antibodies to the proteins indicated on the left. Values are reported as mean ± SD (**B**–**E**). (Key: T1:1 = Time point sampling 1:replicate 1, then onward). (**B**–**E**) Normalized intensities assayed by western blotting for representative proteins that were identified by 2D-DIGE/MS. (T1, 2, 3, and 4 are tissue samplings at weeks 4, 8, 16, and 24, respectively).

**Figure 6 cancers-16-01678-f006:**
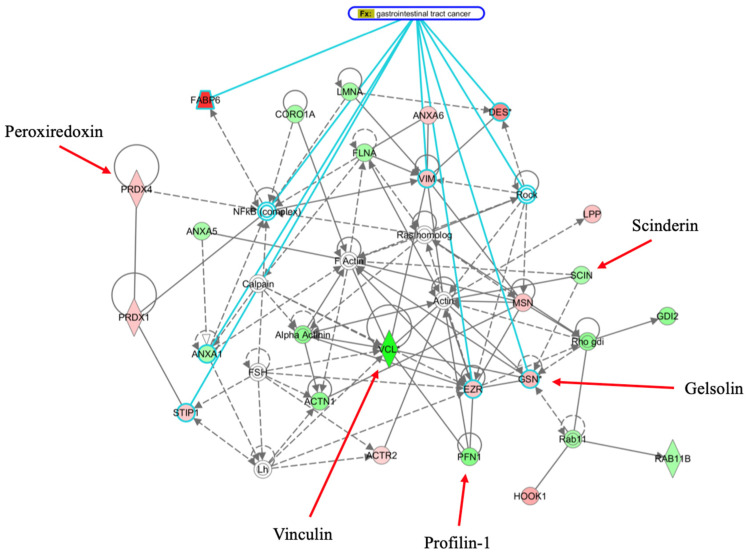
Primary regulatory protein and networks identified, associated with proteins differentially expressed as related to maturation of aberrant crypt foci (ACF) in gastrointestinal tract cancer, when comparing maturing adenoma back to ACF, are shown. The network was generated via the Metacore Integrated Pathway Analysis (IPA) tool using the list of differentially expressed proteins identified by 2-D DIGE/MS analysis (protein identities are shown in Table 2). Individual proteins are represented as nodes, and the different shapes of the nodes represent the functional classes of the proteins. The edges define the relationships of the nodes, and the arrowheads indicate the direction of the interaction. The color of the node indicates activation (green), inhibition (red), and unspecified (clear grey) interactions. Key proteins found in this study to be differentially expressed and highly involved in gastrointestinal cancer networks are labeled.

**Table 1 cancers-16-01678-t001:** Composition of AIN-93G purified diet * used in this study.

Formula		g/Kg
Casein		200.0
L-Cysteine		3.0
Corn Starch		397.486
Maltodextrin		132.0
Sucrose		100.0
Soybean oil		70.0
Cellulose		50.0
Mineral Mix, AIN-93G-MX (04046)		35.0
Vitamin Mix, AIN-93-VX (94047)		10.0
Choline Bitartrate		2.5
TBHQ, antioxidant		0.014
**Macronutrient**	**% dry weight**	**% kcal**
Protein	17.7	18.8
Carbohydrate	60.1	63.9
Fat	7.2	17.2

* Ref: Reeves et al., [24] (Formulated for the growth, pregnancy, and lactational phases of rodents; Cat. #TD.94045; Harlan Teklad Laboratories, Madison, WI, USA).

**Table 2 cancers-16-01678-t002:** Comparisons (protein expression fold-changes) between each time point sampling group (T1, T2, T3, and T4; aberrant crypt foci and adenoma tissue collected from rats sacrificed 4 (control), 8, 16, and 24 weeks after azoxymethane treatment, respectively, with “T4/T1” = T4 (wk 24) vs. T1 (wk 4 or control) time point comparison. Protein identity and gene name, with associated fold-changes and *p*-values between the different time point comparisons are shown.

Gene Name	Protein Identity	Fold Change			One Way ANOVA *p*-Value	*T*-Test *p*-Value		
		T2/T1	T3/T1	T4/T1		T2/T1	T3/T1	T4/T1
*Cps1*	Carbamoyl-phosphate synthase, mitochondrial	−1.21	−1.26	−1.65	0.043	0.24	0.061	0.023
*Flna*	Filamin, alpha isoform	−1.45	−1.68	−1.54	0.02	0.053	0.0015	0.0077
*Vcl*	Vinculin	−2.14	−3.4	−3.89	0.0072	0.034	0.0002	0.0063
*Lrpprc*	Leucine-rich PPR motif-containing, mitochondrial	−2.15	−3.15	−3.55	0.016	0.059	0.0019	0.0012
*Si*	Si 210 kDa protein	−2.46	−3.47	−4.13	0.02	0.044	0.00079	0.0088
*Sf3a1*	Splicing factor 3a, subunit 1	−1.44	−2.14	−2.19	0.03	0.071	0.00048	0.035
*Ganab*	Putative uncharacterized protein	−1.01	1.46	1.24	0.05	0.99	0.065	0.16
*Actn1*	Actinin	−1.53	−1.47	−1.91	0.041	0.17	0.21	0.049
*Hook1*	Hook homolog 1 (predicted)	1.38	1.7	1.99	0.0029	0.041	0.0052	0.0096
*Gsn*	Isoform 2 of gelsolin	1.16	1.35	1.59	0.044	0.12	0.093	0.014
*Immt*	Inner membrane protein, mitochondrial (predicted)	1.11	1.19	1.68	0.0096	0.014	0.071	0.018
*Scin*	Scinderin	−1.13	−1.38	−1.48	0.028	0.44	0.04	0.016
*Lpp*	Lipoma-preferred partner homolog	1.27	1.3	1.47	0.03	0.062	0.077	0.022
*Ezr*	Ezrin	1.1	1.56	1.32	0.033	0.13	0.0064	0.16
*Lmna*	Lamin-A	−1.52	−1.92	−1.55	0.028	0.048	0.029	0.071
*Tf*	Isoform 1 of serotransferrin	−1.58	−1.89	−1.86	0.039	0.028	0.011	0.054
*Clca3*	Putative uncharacterized protein	1.06	−1.36	−1.56	0.046	0.58	0.044	0.1
*Stip1*	Stress-induced-phosphoprotein 1	1.15	1.5	1.25	0.0028	0.11	0.0031	0.021
*Yars*	Tyrosyl-tRNA synthetase, cytoplasmic	1.23	1.73	1.4	0.025	0.14	0.017	0.061
*Coro1a*	Coronin-1A	−1.19	1.09	−1.55	0.022	0.18	0.65	0.029
*Vim*	Vimentin	1.2	1.45	1.41	0.012	0.018	0.0029	0.038
*Des*	Desmin	1.37	1.61	2.76	0.025	0.07	0.016	0.04
*Sept11*	Isoform 1 of septin-11	1.02	1.38	1.42	0.012	0.88	0.036	0.024
*Eno1*	Alpha-enolase	−1.55	−1.85	−1.89	0.043	0.056	0.012	0.024
*Oat*	Ornithine aminotransferase, mitochondrial	−1.37	−1.67	−1.71	0.0018	0.044	0.011	0.0098
*Acy1*	Aminoacylase-1A	−1.5	−1.92	−1.72	0.012	0.069	0.00071	0.0052
*Inpp1*	Inositol polyphosphate-1-phosphatase	−1.42	−1.62	−1.63	0.031	0.072	0.00036	0.0088
*Acads*	Acyl-Coenzyme A dehydrogenase	1.11	1.12	1.49	0.035	0.39	0.21	0.034
*Mdh1*	Malate dehydrogenase, cytoplasmic	−1.33	−1.41	−1.48	0.02	0.049	0.025	0.023
*Clic4*	Chloride intracellular channel protein 4	1.48	1.38	1.58	0.034	0.062	0.056	0.031
*Prdx4*	Peroxiredoxin-4	1.18	1.51	1.42	0.02	0.056	0.016	0.028
*Prdx1*	Peroxiredoxin-1	1.03	−1.11	1.35	0.04	0.67	0.017	0.12
*Myl9*	Myosin, light polypeptide 9, regulatory	1.35	1.25	1.68	0.013	0.0075	0.064	0.016
*Pfn1*	Profilin-1	−1.39	−1.76	−2.08	0.015	0.066	0.00067	0.02
*Fabp6*	Gastrotropin	8.08	5.54	4.88	0.012	0.025	0.0046	0.0023

## Data Availability

The original contributions and data on which the findings in this study are based are included within the article, supporting material, and at: JPost Repository (ID = JPST003023; https://repository.jpostdb.org/entry/JPST003023; (accessed on 8 April 2024) and ProteomeXchange ID = PXD051252; https://proteomecentral.proteomexchange.org/cgi/GetDataset?ID=PXD051252, accessed on 8 April 2024).

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
