# Peer review of "Proteins Involved in Focal Cell Adhesion and Podosome Formation Are Differentially Expressed during Colorectal Tumorigenesis in AOM-Treated Rats"

_cancers, 2024, doi:10.3390/cancers16091678_

Round 1

Reviewer 1 Report

Comments and Suggestions for Authors

NICE WORK!AN EXCELLENT APPROACH TO NEW DIAGNOSTIC OPTIONS.THE ONLY THINGS TO ADD ARE :THE MOLECULAR PROFILE(51-59)ITS ALMOST SIMILAR WITH OTHER CANCERS(LUNG,BREAST...)PLEASE EXPLAIN WHY FOR COLON CA WE SHOULD ASPECT SOMETHING DIFFERENT AS BIOMARKER.SECONDLY  CANCEROUS EVENTS ARE DAILY ROUTINE,(WE DONT GO TO CANCER EASILY BECOSE OF THE QUIQ TURNOVER OF INTESTINAL CELLS,)FOR CERTAIN YOU SHOULD CHECK NORMAL TISSUE TO SEE THE CONDITION OF THE G.I. OF THE SAME UNIT.

Reviewer 2 Report

Comments and Suggestions for Authors

Swain and kresak presented a research titled, "From aberrant crypt to adenoma: Changes in the expression of regulatory proteins, including those involved in focal cell adhesion and cell migration, during early colorectal tumorigenesis in an AOM murine model"

The authors studied Protein expression profiles of azoxymethane (AOM)-induced F344 rat colon ACF and adenomas were compared at four time points. By determining protein expression changes in ACF in proteins associated with adenocarcinoma, authors suggested the future  identification of biomarker to understand the molecular biology of intestinal tumorigenesis. 

Following concerns need to be addressed by authors:-

1. Title is too long, a concise title to be given to represent the significance of the work.

2. Plagiarism detected in introduction part (Initial two paragraphs) and most of the experimental sections (2.4 to 2.8) . Authors need to address them more seriously. If authors using similar protocols as like their previous papers, then, they can cite the paper in the protocol section. If they had some modification, then, they can ,mention that modified protocol from our earlier protocol has been used and cite properly.

3. Did the authors determined the standard deviation (SD) or standard error of mean (SEM) for Figure 5A-E. In 5E, the SEM is high for T1, T3, T4 . Authors need to clarify this point.

Reviewer 3 Report

Comments and Suggestions for Authors

The manuscript by Swain and Kresak investigates the protein expression changes within ACF and adenoma in AOM-induced rat colon cancer model.  The analyses identified several proteins that seemed to be differentially expressed within these lesions.  Characterization of protein expressions associated with various phases of cellular transformation is important to understand the mechanisms of progression and to develop effective therapeutic strategies.  The study design is straight forward, and analyses are presented in detail, however there are some data that should be included in the manuscript.  As a minor comment, the title should be changed to describe the study and findings.  It reads like a review article.

Specific comment:

-       ACF can develop any time during T1-T3 (or T4).  Even though the frequency of ACF decreases, some of them may be at early phase of development in the later stages. Are there any specifications for harvesting particular ACF/adenomas? 

-       Quantification and size of lesions collected at each time point should be included.

-       Adding some histology figures may strengthen the manuscript.

-       Some of the information in the Discussion is too descriptive that they should be reported in the Result section.  

Comments on the Quality of English Language

Some editing of English language required.

Round 2

Reviewer 3 Report

Comments and Suggestions for Authors

The authors addressed reviewers' comments and the revised manuscript includes more details.  It is still unclear whether comparing ACF from different time points (of similar sizes) can be considered as an assessment of "progression".  
